# An extrafollicular pathway for the generation of effector CD8+ T cells driven by the proinflammatory cytokine, IL-12

**Suhagi Shah[1], Gijsbert M Grotenbreg[2†], Amariliz Rivera[1], George S Yap[1]\***

[1]Center for Immunity and Inflammation, New Jersey Medical School, Rutgers University, Newark, United States; [2]Immunology Programme, Departments of Microbiology and Biological Sciences, National University of Singapore, Singapore, Singapore

**Abstract** The proinflammatory cytokine IL-12 drives the generation of terminally differentiated KLRG1+ effector CD8+ T cells. Using a *Toxoplasma* vaccination model, we delineate the sequence of events that naïve CD8+ T cells undergo to become terminal effectors and the differentiation steps controlled by IL-12. We demonstrate that direct IL-12 signaling on CD8+ T cells is essential for the induction of KLRG1 and IFN-γ, but the subsequent downregulation of CXCR3 is controlled by IL-12 indirectly through the actions of IFN-γ and IFN-γ-inducible chemokines. Differentiation of nascent effectors occurs in an extrafollicular splenic compartment and is driven by late IL-12 production by DCs distinct from the classical CD8α+ DC. Unexpectedly, we also found extensive proliferation of both KLRG1− and KLRG1+ CD8+ T cells in the marginal zone and red pulp, which ceases prior to the final KLRG1Hi CXCR3Lo stage. Our findings highlight the notion of an extrafollicular pathway for effector T cell generation.

\*For correspondence: yapgs@ njms.rutgers.edu

Present address: †121 Bio LLC, Cambridge, United States

Competing interests: The authors declare that no competing interests exist.

## Introduction

The activation of naïve CD8+ T cells occurs in the T cell areas of secondary lymphoid organs (*Mempel et al., 2004*). Once activated, CD8+ T cells are thought to undergo rapid clonal expansion and differentiate into IFN-γ-producing effector cytotoxic lymphoid cells (CTLs) through an 'autopilot' mechanism (*Mercado et al., 2000*; *Bevan and Fink, 2001*; *Kaech and Ahmed, 2001*; *van Stipdonk et al., 2001*). Consistent with this paradigm, CD8α+ dendritic cells (DCs), which are resident in the splenic white pulp, are critical for both antigen cross-presentation (*den Haan et al., 2000*) and production of a key pro-inflammatory cytokine, IL-12 (*Reis e Sousa et al., 1997*; *Mashayekhi et al., 2011*). IL-12 is an important signal 3 cytokine that drives clonal proliferation of activated CD8+ T cells in vitro (*Curtsinger et al., 1999*) as well as their effector cytokine production in vivo (*Schmidt and Mescher, 1999, 2002*; *Valenzuela et al., 2002*; *Liu et al., 2006*; *Wilson et al., 2008*). However, following their activation and migration into the inner PALS (*Reis e Sousa et al., 1997*), CD8α+ DCs become hyporesponsive and cease IL-12 production (*Reis e Sousa et al., 1999*), raising the question of whether this initial early burst of IL-12 is sufficient to drive end-stage effector CD8+ T cell differentiation to completion.

Recently, there has been mounting evidence that additional events occurring outside the T cell area are critical for primary TH1 and CTL effector differentiation. CXCR3 deficiency results in an inefficient generation of cytokine-producing effector cells and instead favors a response skewed towards memory cells (*Kohlmeier et al., 2011*; *Kurachi et al., 2011*; *Groom et al., 2012*). CXCR3-mediated outmigration of T cells into the splenic marginal zone (MZ) (*Kurachi et al., 2011*) or the peripheral medullary areas of lymph nodes (*Groom et al., 2012*) enables interactions with

**eLife digest** The immune system helps to protect us from cancer, infection by microbes and other diseases. There are several different types of immune cells that each have particular roles. For example, cytotoxic T cells can kill other cells in the body that are damaged or infected. These cells are found in various locations around the body—including a region of the spleen known as the white pulp—where they wait in an inactive state until they detect signals from a damaged or infected cell. These T cells divide and mature to produce populations of active T cells known as effector cytotoxic lymphoid cells (or CTLs for short), a process which is thought to occur within the white pulp.

A small protein called cytokine IL-12 is involved in the production of CTLs. The cytokine is released from other immune cells and causes the activated T cells to divide and mature. It has long been believed that IL-12 produced in the white pulp early on in the process is sufficient to drive this process, but more recent work suggests that sustained production of IL-12 in other areas of the spleen that are accessible to the bloodstream may be needed.

Here, Shah et al. studied the generation of cytotoxic T cells in mice that had been exposed to a vaccine against a disease called Toxoplasmosis. Their experiments show that IL-12 drives both the early and late stages of CTL production. In the early stages, the T cells respond to IL-12 that is secreted by a group of 'lymphoid dendritic' cells in the white pulp. However, in the later stages, the T cells move away from the white pulp to other parts of the spleen known as the marginal zone and red pulp, where a distinct group of 'myeloid dendritic' cells also produce IL-12 and direct the final maturation of the CTLs.

Shah et al.'s findings also show that the process in which cytotoxic T cells divide and later mature to produce CTLs involves a series of tightly controlled events that mostly occur outside of the white pulp. These observations provide a new perspective on how to develop vaccines and other treatments that more efficiently generate the CTLs needed to protect against infections and cancer.

IL-12 producing myeloid cells. The role of CXCR3 in lymphocyte peripheralization also extends to the memory response and is essential for secondary effector T cell generation from central memory T cells (*Sung et al., 2012*; *Kastenmuller et al., 2013*). These recent findings argue that an initial exposure to IL-12 and other signal 3 cytokines during the early stages of T cell activation may not be sufficient for effector T cell differentiation. Instead, a subsequent exposure to IL-12 at extrafollicular sites appears to drive effector CD8+ T cell differentiation to completion. It is also possible that this requirement may stem from the initial lack of a robust CD8α+ DC- derived IL-12 during viral and bacterial immune responses (*Dalod et al., 2003*; *Edelson et al., 2011*).

During CD8+ T cell priming, the inflammatory milieu governs effector CD8+ T cell generation. In particular, the cytokine IL-12 potently promotes CD8+ T cell effector differentiation and function, through T-bet-dependent induction of KLRG1 expression and IFN-γ production (*Joshi et al., 2007*; *Wilson et al., 2008*, *2010*). In addition to these effects, IL-12 appears to also control the migratory potential of primary and effector T cells, by negatively regulating CXCR3 expression (*Slutter et al., 2013*). Given the possibility of IL-12 being produced both within the T cell area and at extrafollicular sites, it remains unclear how IL-12 temporarily and spatially controls these key facets of the primary effector CTL response. Furthermore, although it is assumed that these IL-12 effects are CD8+ T cell-autonomous, the possibility exists that regulation may be indirectly mediated through the actions of this cytokine on other IL-12 responsive immune cells.

In order to address these questions, we used a *Toxoplasma gondii* vaccination model to conduct a detailed analysis of the role of IL-12 in the generation, function and migratory potential of parasite-specific effector CD8+ T cells. We had previously identified this *tgd057*-reactive H-2Kb-restricted CD8+ T cell response in mice vaccinated with an attenuated, uracil auxotrophic strain of *T. gondii* (CPS), known to elicit CD8+ T cell-dependent protective immunity (*Fox and Bzik, 2002*) and have demonstrated a strict in vivo requirement for IL-12 to generate KLRG1+ effector CTLs (*Wilson et al., 2008*, *2010*). Our results reveal that the sequence of differentiative events that culminate in the production of primary end-stage effector CD8+ T cells occurs over a protracted period and that IL-12 exerts regulatory functions at both early and late phases of effector cell generation. The effects of IL-12 in upregulating KLRG1 expression and priming for IFN-γ production

require CD8$^+$ T cell intrinsic cytokine signaling. In contrast, we found that the belated downregulation of CXCR3 on effector CD8$^+$ T cells is indirectly regulated by IL-12 and is instead controlled by a pathway in which IFN-γ and IFN-γ-inducible chemokines mediate this downmodulation. Using an in vivo intravascular staining method (*Olson et al., 2013*; *Anderson et al., 2014*), we were able to reveal that these later stages of effector CD8$^+$ T cell differentiation occur extrafollicularly, involving DCs as cellular sources of both non-CD8α$^+$ DC-derived IL-12 and CXCR3-ligands. Surprisingly, we also found extensive proliferation of both KLRG1$^-$ and KLRG1$^+$ CD8$^+$ T cells in the MZ and red pulp (RP). Taken together with earlier studies (*Lauvau et al., 2001*; *Cockburn et al., 2010*), our findings argue against the notion that effector CTL generation occurs through an 'autopilot' sequence and, instead, involves a multi-leveled progression of effector T cell precursors through distinct splenic microenvironments, where their differentiation is controlled by a complex interplay with locally positioned activated immune cells.

## Results

### CD8$^+$ T cell proliferative response is IL-12 independent while effector cell differentiation is IL-12 dependent

To determine the early effects of IL-12 on CD8$^+$ T cell proliferation and differentiation during *T. gondii* infection, we used a tetramer-based enrichment method (*Klenerman et al., 2002*; *Moon et al., 2007*) to enumerate H-2K$^b$-restricted CD8$^+$ T cells specific for the *T. gondii* antigen *tgd057* (*Wilson et al., 2010*) in wild-type (WT) and IL-12p35 deficient hosts following CPS vaccination. The tetramer-based enrichment method allows for a ~2-log increase in detection of *tgd057*-specific CD8$^+$ T cells (*Figure1—figure supplement 1*). To accurately enumerate the exact numbers of *tgd057*-specific CD8$^+$ T cells when their frequencies are very low we mixed a known number of naïve Thy1.1$^+$-marked *tgd057*-specific CD8$^+$ T cells from a somatic cell nuclear transfer (SCNT) mouse (*Kirak et al., 2010a*) with spleen cells from individual naïve mice (*Figure 1—figure supplement 2*). We are able to detect 2093 ± 273 and 1200 ± 138 endogenous *tgd057*-specific CD8$^+$ T cells in spleens of naïve WT and IL-12p35 deficient mice, respectively (*Figure 1—figure supplement 2*). *Figure 1A* shows that there is little change in absolute cell numbers of *tgd057*-specific CD8$^+$ T cells between day 0 and day 4 post-vaccination in the spleen. However, we can detect a 10-fold clonal expansion occurring between days 4 and 5-post infection in spleens of both WT and IL-12 deficient mice. The absolute numbers of *tgd057*-specific CD8$^+$ T cells continue to increase at similar rates through day 7 in WT and IL-12 deficient mice. These results indicate that the lack of IL-12 signals does not affect the timing or the magnitude of the proliferative response of *tgd057*-specific CD8$^+$ T cells during *T. gondii* vaccination.

Despite the apparent lack of a role for IL-12 in the proliferative recruitment of *tgd057*- specific CD8$^+$ T cells, IL-12 could play a crucial role in the early diversification of CD8$^+$ T cells into end-stage effector CD8$^+$ T cells vs memory precursor type cells. Indeed, previous studies in our laboratory and others have shown that IL-12 is crucial for the differentiation of KLRG1$^+$ effector CTLs through T-bet dependent induction of effector genes (*Joshi et al., 2007*; *Wilson et al., 2008*, *2010*). To determine how IL-12 signals affect the early differentiation of *tgd057*-specific CD8$^+$ T cells, we monitored the expression of CD62L and KLRG1 following *T. gondii* vaccination. We have previously used these cell surface markers to define four specific CD8$^+$ T cell stages: F1 (T$_{CM}$: CD62L$^+$, KLRG1$^-$), F2 (T$_{EM}$: CD62L$^-$, KLRG1$^-$), F3 (T$_{EFF}$: CD62L$^-$, KLRG1$^+$) and F4 (CD62L$^+$, KLRG1$^+$) (*Wilson et al., 2008*, *2010*). While F1 and F2 are determined to be central and effector memory CD8$^+$ T cells, respectively; F3 are the late stage highly IFN-γ-producing effector CD8$^+$ T cells, little is still known about the phenotype and function of the F4 stage CD8$^+$ T cells. We do not observe consistent changes in CD8$^+$ T cell stage distribution until day 3 in either WT or IL-12 deficient hosts (*Figure 1B*). However, by day 3, ~7–8% of the *tgd057*-specific CD8$^+$ T cells start to commit to putative T$_{EM}$ and T$_{EFF}$ cells by downregulating CD62L and increasing KLRG1 expression in the WT hosts (*Figure 1B*). In IL-12p35 deficient hosts, L-selectin downregulation is also observed starting on day 3, however KLRG1 upregulation is absent (*Figure 1B*). On day 4, there is a dramatic increase in the frequency of *tgd057*-specific CD8$^+$ T cells that have downregulated CD62L, a third of which also acquire KLRG1 in WT mice. However, in the absence of IL-12, there is an attenuation of KLRG1 expression resulting in an even larger frequency of F2 *tgd057*-specific CD8$^+$ T cells (*Figure 1B*). The frequency of KLRG1$^+$ *tgd057*-specific CD8$^+$ T cells continues to increase through day 7 post-vaccination in WT hosts; yet, in IL-12 deficient

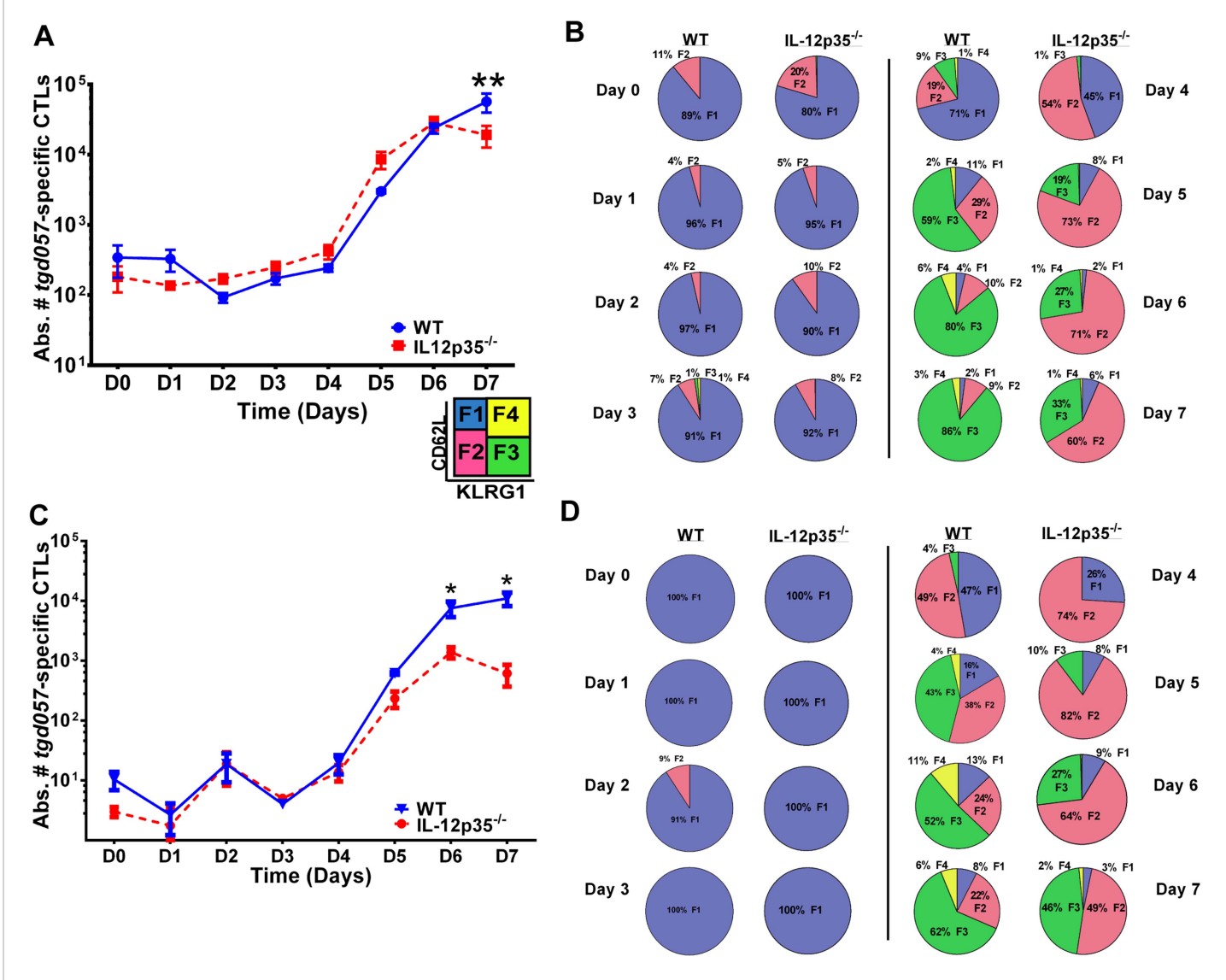

**Figure 1**. CD8+ T cell proliferative response is IL-12 independent while effector cell differentiation is IL-12 dependent. (**A**) Absolute numbers of *tgd057*-specific CD8+ T cells in spleen after CPS vaccination in wild-type (WT) and *IL-12p35−/−* mice. (**B**) *tgd057*-specific CD8+ T cells were analyzed for CD44, CD62L and KLRG1 cell surface expression. Pie charts represent frequency averages of F1 (CD62L+ KLRG1−), F2 (CD62L−, KLRG1−), F3 (CD62L−, KLRG1+), and F4 (CD62L+, KLRG1+) subsets of *tgd057*-specific CD8+ T cells from WT and *IL-12p35−/−* spleens D0–D7 post CPS vaccination. Data shown include only CD44hi cells on D4–D7 and include all *tgd057*-specific CD8+ T cells on D0–D3. (**C**) Absolute numbers of *tgd057*-specific CD8+ T cells in peritoneal exudate cells (PECs) of WT and *IL-12p35−/−* mice after CPS vaccination. Cells were characterized based on *tgd057*-specific CD8+ T cells. (**D**) Average of frequencies of F1 (CD62L+ KLRG1−), F2 (CD62L−, KLRG1−), F3 (CD62L−, KLRG1+), and F4 (CD62L+, KLRG1+) subsets of *tgd057*-specific CD8+ T cells in PECs D0–D7. Data represent 5 independent experiments with 3–5 mice per group per experiment. Data were analyzed using two-way ANOVA and (**A**) Holms-Sidak or (**C**) Bonferroni post-hoc tests; *$p < 0.05$. See *Figure 1—figure supplements 1, 2* for the methodology employed to enumerate absolute numbers of *tgd057*-specific CD8+ T cells.

The following figure supplements are available for figure 1:

**Figure supplement 1**. Tetramer-based enrichment increases detection of epitope-specific naïve CD8+ T cells in spleen.

**Figure supplement 2**. Accurate determination of endogenous naïve *tgd057*:H-2Kb-specific CD8+ T cells.

mice, KLRG1 upregulation remains attenuated and the frequencies of F3 stage *tgd057*-specific CD8[+] T cells do not reach levels seen WT hosts (*Figure 1B*). These results indicate that IL-12 signaling has an important role on the differentiation of KLRG1[+] effector *tgd057*-specific CD8[+] T cells, and that this occurs earlier than the proliferative burst seen after day 4. In contrast, downregulation of CD62L and subsequent clonal expansion are IL-12 independent.

## During effector CD8[+] T cell differentiation, expression of CXCR3 is downregulated in an IL-12 dependent manner

In spite of the fact that IL-12 signals did not play a crucial role in proliferation of *tgd057*-specific CD8[+] T cells in the spleen, we observed that case to be different at the site of infection. We enumerated *tgd057*-specific CD8[+] T cells in the peritoneal exudate cells (PECs) using the SVLAFRRL:H-2K[b] tetramer (*Figure 1C*). Surprisingly, we found that the absolute numbers of *tgd057*-specific CD8[+] T cells in the PECs in an IL-12 deficient environment by day 6 were attenuated by one log in absolute numbers as the height of the adaptive immune response was reached (*Figure 1C*).

The observation that *tgd057*-specific CD8[+] T cells traffic to the peritoneum is attenuated when IL-12 signals are lacking prompted us to investigate the role of IL-12 for effector *tgd057*-specific CD8[+] T cells migration. The T-box protein, T-bet, is a master regulator of the differentiation and functional activity of effector CD8[+] T cells. It has been shown to associate with the promoters of genes for KLRG1 (*Joshi et al., 2007*), IFN-γ (*Joshi et al., 2007*), as well as the chemokine receptor, CXCR3 (*Beima et al., 2006*; *Harms Pritchard et al., 2015*). We hypothesized that since KLRG1 expression and the production of IFN-γ by F3 late stage effector CD8[+] T cells is IL-12 dependent (*Wilson et al., 2008*, *2010*), CXCR3 expression may also be IL-12 dependent, possibly explaining the attenuation of effector cells to the site of infection in our model. Analysis of CXCR3 expression as CD8[+] T cells differentiate from F1 ($T_{CM}$) to F2 ($T_{EM}$) cells in the spleen show that CXCR3 expression is highly expressed when CD8[+] T cells are CD62L[+] or CD62L[−] (*Figure 2A,B*). Interestingly, our data demonstrate that the early expression of CXCR3 is IL-12 independent, but T-bet dependent (*Figure 2A,B*) and as these cells differentiate into end-stage effector IFN-γ producing KLRG1[+] CD8[+] T cells, they downregulate CXCR3 (*Figure 2A,B*), indicating that IL-12 has late downregulatory effects on CXCR3 expression.

## Downregulation of CXCR3 expression on CD8[+] T cells occurs following KLRG1 induction in the splenic RP

We next wished to determine where in the spleen KLRG1 upregulation and CXCR3 down-regulation occurs as *tgd057*-specific CD8[+] T cells differentiate into late-stage effectors. In order to study the white pulp and RP distribution of the CD8[+] T cells within the spleen, we intra-venously injected mice with a fluorescently conjugated anti-CD8α antibody on days 4 and 7 post CPS vaccination (*Figure 3A*). Tissue was isolated after a brief delay to quickly label only cells exposed to the blood and RP within the spleen, but excluded cells in white pulp (*Olson et al., 2013*; *Anderson et al., 2014*). Interestingly, our results suggest that on day 4, *tgd057*-specific CD8[+] T cells upregulate KLRG1 expression after trafficking to the RP (*Figure 3B*). By day 7 post-vaccination, the *tgd057*-specific CD8[+] T cells are found to be equally localized to the white pulp and the RP prior to increasing their KLRG1 expression, where, once KLRG1 expression is upregulated, the majority (∼80%) of the cells is located in the RP. Thus, the early upregulation of KLRG1 expression on *tgd057*-specific CD8[+] T cells occurs mainly in the RP, where they may be encountering a secondary signal of the pro-inflammatory cytokine, IL-12, for later dif-ferentiation into end-stage effector CD8[+] T cells. Because the preferential localization of CD62L[−] KLRG1[−] and CD62L[−] KLRG1[+] *tgd057*-specific CD8[+] T cells is in the RP on day 4, we next determined whether CXCR3 is concomitantly downregulated as KLRG1 expression is increased. As *Figure 3C* indicates, at day 4 CXCR3 expression in both KLRG1[−] and KLRG1[+] CD8[+] T cells is not substantially decreased, but by day 7, the KLRG1[+] CD8[+] T cells which are predominantly RP-localized have completely downregulated CXCR3. Taken together, these results demonstrate that prior to the upregulation of KLRG1, effector precursor CD8[+] T cells expressing high levels of CXCR3 can migrate to the RP, where they upregulate KLRG1 and then downregulate CXCR3 as they differentiate into late-stage effector CD8[+] T cells.

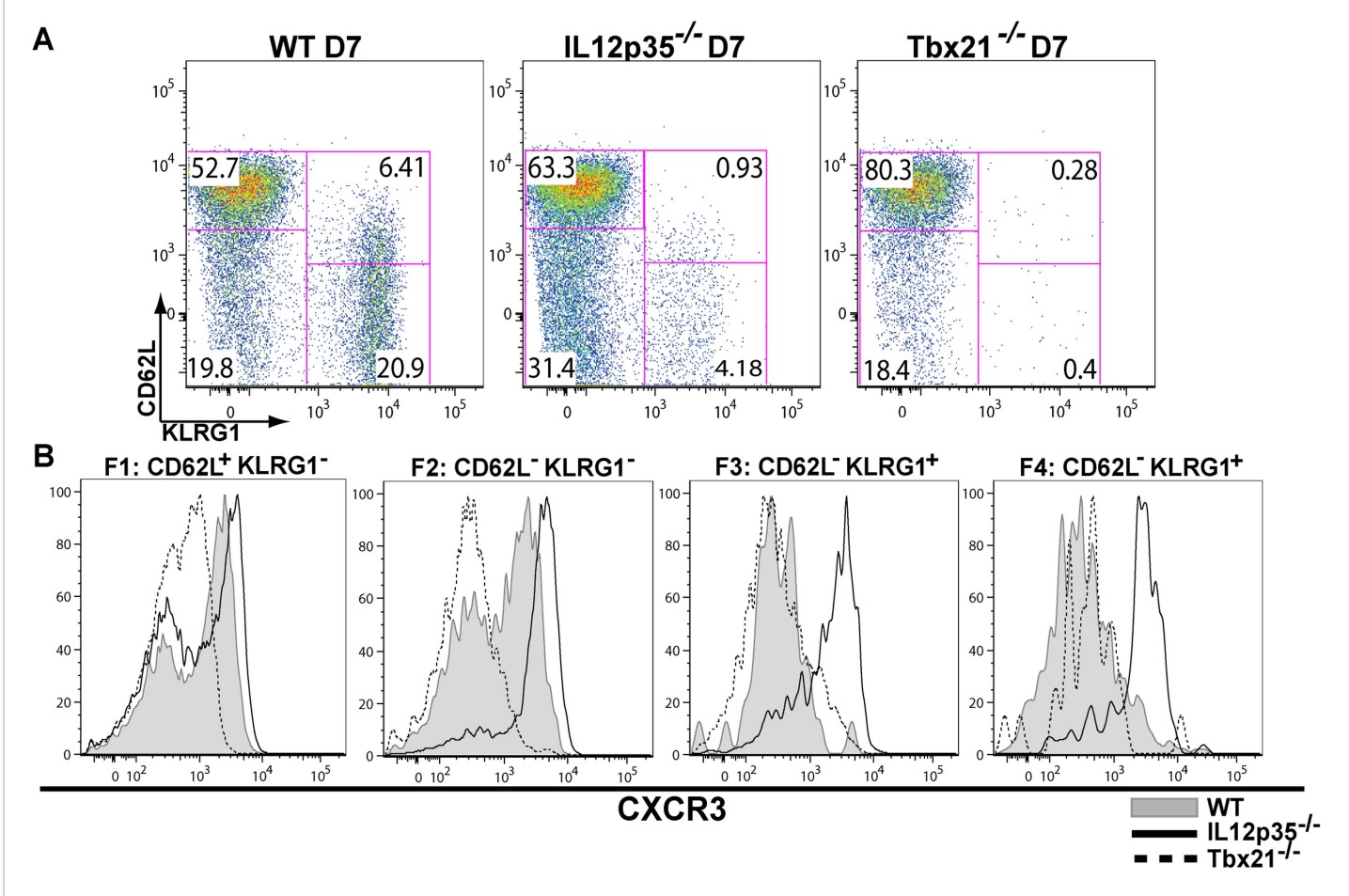

**Figure 2**. During effector CD8+ T cell differentiation, expression of CXCR3 is downregulated in an IL-12 dependent manner. CD62L and KLRG1 surface expression (**A**) and CXCR3 surface expression (**B**) on total CD8+ T cells were assessed by flow cytometry 7 days post-vaccination. Data represent 3–4 independent experiments with 4–5 mice per group per experiment. Mean ± SEM, data were analyzed using unpaired t test, and Holms-Sidak post-hoc test, *p ≤ 0.05, **p ≤ 0.01, ***p ≤ 0.001.

## Late effects of IL-12 on the CD8+ T cell differentiation, function and chemokine receptor expression

Our data thus far suggest that IL-12 exerts early effects on the differentiation of effector *tgd057*-specific CD8+ T cells during *T. gondii* infection by upregulating KLRG1 even prior to clonal expansion (*Figure 1*), but also plays a later role in the downregulation of CXCR3 expression (*Figure 2*). IL-12 may be produced only early during vaccine priming and programs the entire effector differentiation pathway over time. Alternatively, IL-12 may be produced at both early and late time points, potentially by distinct APCs. To address the latter scenario, we neutralized IL-12 late (D3) and compared its effects on CD8+ T cell differentiation to early (D0) and continuous neutralization (D0 and D3) following CPS vaccination. Consistent with our earlier results, the blockade of IL-12 at early, late or both time points did not affect absolute *tgd057*-specific CTL numbers (*Figure 4A*). When IL-12 is neutralized only on day 3, KLRG1 expression and IFN-γ-production were still attenuated, albeit less markedly compared to the effects of early and continuous blockade of the cytokine (*Figure 4B,C*). Importantly, we find that late neutralization of IL-12 interrupts the downregulation of CXCR3 expression on late-stage effector *tgd057*-specific CD8+ T cells (*Figure 4D*). These results confirm that IL-12 is required for early programming of effector differentiation, and indicate that IL-12 production occurs days after initial CPS vaccine exposure and continues to induce further differentiation of effector-fated CD8+ T cells.

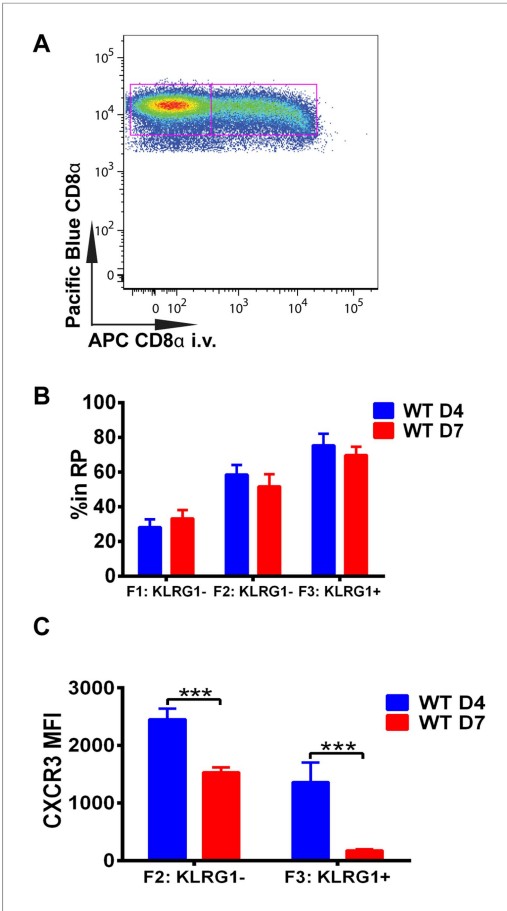

**Figure 3**. Downregulation of CXCR3 expression on CD8+ T cells occurs following KLRG1 induction in the splenic red pulp (RP). (**A**) Representative FACS profile identifying splenic RP and white pulp by differential staining with i.v injected APC conjugated anti-CD8α antibody. (**B**) Compiled data of RP distribution of F1, F2 and F3 *tgd057*-specific CD8+ T cells in spleen on D4 and D7 post CPS vaccination. (**C**) CXCR3 expression on F2-KLRG1− and F3-KLRG1+ *tgd057*-specific CD8+ T cells on D4 and D7 post CPS vaccination. Data are representative of 3 independent experiments with 6–8 mice per experiment. Mean ± SEM, data were analyzed using unpaired t test, and Holms-Sidak post-hoc test, ***p ≤ 0.001.

## IL-12 mediates surface CXCR3 downregulation in a CD8+ T cell-extrinsic manner

The observation that IL-12-mediated downregulation of CXCR3 can be temporally dissociated from its effects on KLRG1 upregulation (*Figure 2B,C*), which was previously shown to be a CD8+ T cell-autonomous effect of the cytokine (*Slutter et al., 2013*), prompted us to ask if the IL-12 effects are CD8+ T cell-intrinsic or extrinsic. In addition, CXCR3 downregulation on KLRG1+ CD8+ T cells occurs in the RP (*Figure 3B,C*), where IL-12 producing cell types other than CD8α+ DCs may be involved. Specifically, the late-acting IL-12 may be secreted by inflammatory-monocyte derived DCs, and has been shown to require NK (*Goldszmid et al., 2012*) or T_H1-CXCR3+ CD4+ T cell -derived IFN-γ (*Cohen et al., 2013*) for DC activation. This scenario raises the possibility that other cells besides the developing effector CD8+ T cells may be the targets of IL-12 signaling. In order to determine if IL-12 mediates CXCR3 downregulation directly, we adoptively transferred a 1:1 mixture of naïve WT and IL-12rβ2 deficient monoclonal *tgd057*-specific CD8+ T cells (derived via SCNT cloning) into a CD45.1+ WT B6 host (*Figure 5A*). 7 days after CPS-vaccination, we observe that the lack of IL-12 signaling does not alter the frequency of IL-12rβ2 deficient donor CD8+ T cells (*Figure 5B*), but CXCR3 downregulation is inhibited (*Figure 5C*). These results confirm earlier conclusions that IL-12 is not essential for T cell expansion in vivo and are consistent with the notion that IL-12 directly signals the downregulation of CXCR3 expression. However, when we further compared CXCR3 expression between KLRG1− vs KLRG1+ stages, we found that the F3 stage KLRG1+ CD8+ T cells had downregulated CXCR3 regardless of their ability to signal IL-12 (*Figure 5D*). We conclude that CXCR3 is downregulated in a non-CD8+ T cell autonomous manner, which is not consistent with previous results (*Slutter et al., 2013*). The apparent lack of CXCR3 downregulation is due to the predominant deficiency in KLRG1 expression on the IL-12Rβ2 deficient CD8+ T cells (*Figure 5E*). Unlike CXCR3, KLRG1 expression and IFN-γ production, that are known to be directly regulated by IL-12 signaling, behave in a cell autonomous manner (*Figure 5E,F*).

## An alternative pathway involving IFN-γ and IFN-γ-inducible chemokines mediates suppression of CXCR3 on late-stage effector CD8+ T cell precursors

To explore the mechanism of CD8+ T cell-extrinsic downregulation of CXCR3 expression, we considered the role of IFN-γ because it promotes the expression of CXCR3 chemokine ligands

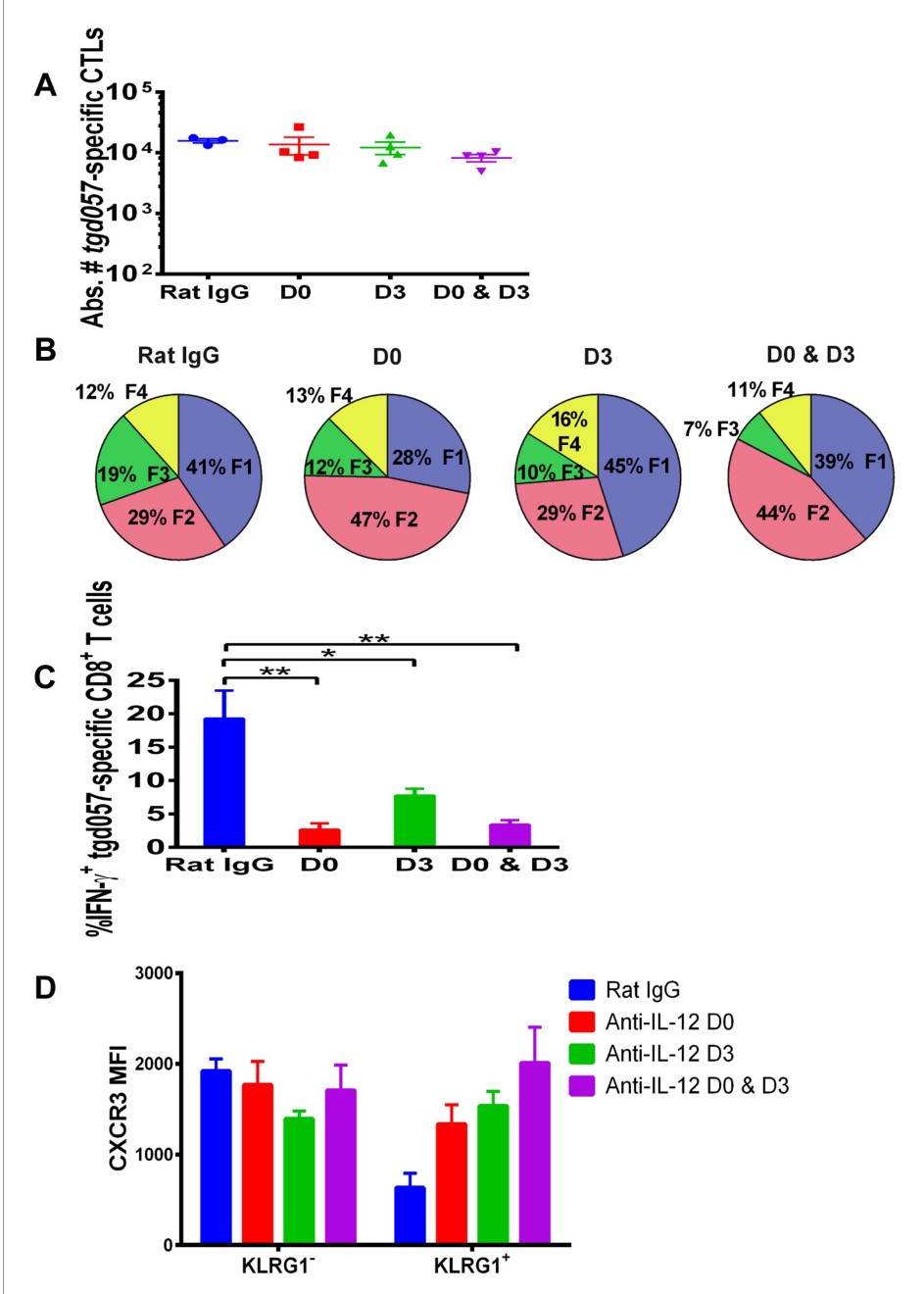

**Figure 4**. IL-12 exerts both early and late effects on CD8+ T cell differentiation, function and chemokine receptor expression. Anti-IL-12p40 mAb was administered i.p. on D0, D3 or both days post CPS vaccination. Spleens were harvested on D5 post CPS vaccination. (**A**) Absolute numbers of *tgd057*-specific CD8+ T cells, (**B**) average frequency of F1–F4 stages defined by cell surface expression of CD62L and KLGR1, (**C**) IFN-γ production and (**D**) CXCR3 downregulation on *tgd057*-specific CD8+ T cells. Data are from 3 independent experiments with 3–4 mice per group per experiment. Mean ± SEM, data were analyzed using multiple unpaired t test, and Holm-Sidak post-hoc test, *p ≤ 0.05, **p ≤ 0.01.

and because IL-12 may be acting indirectly through IFN-γ production by non-CD8+ T cells. To first test if blocking IFN-γ alone can prevent the downregulation of CXCR3 on CD8+ T cells in vivo, we neutralized IFN-γ late (D3) during CPS vaccination. We found that in vivo blockade of IFN-γ as late as D3 disrupts the downregulation of CXCR3 expression in KLRG1-expressing

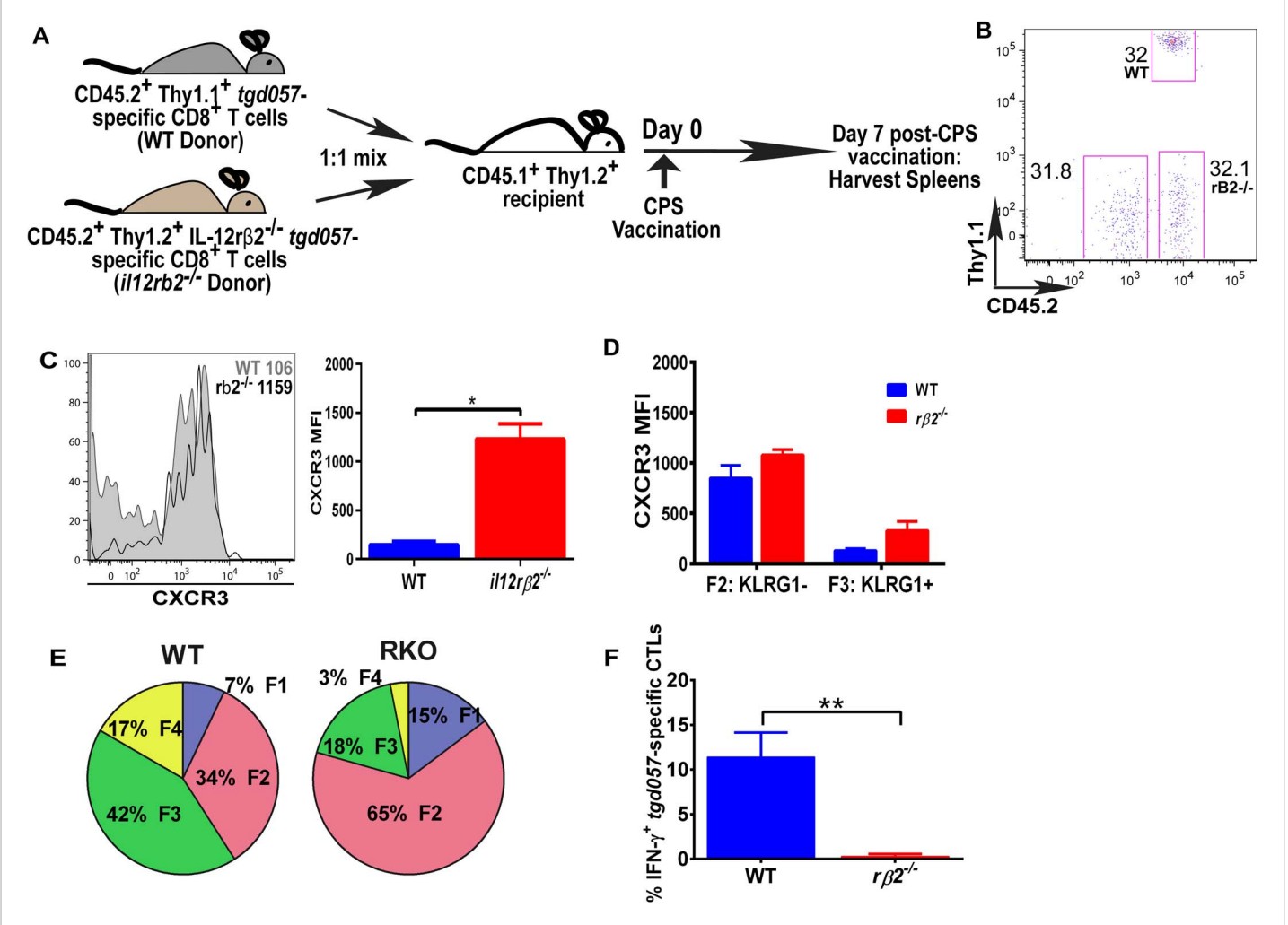

**Figure 5**. IL-12 mediates CXCR3 downregulation in a CD8+ T cell-extrinsic manner. (**A**) A schematic representation of adoptive co-transfer of naïve 500 WT somatic cell nuclear transfer (SCNT, CD45.2+, Thy1.1+) and naïve 500 *il12rβ2−/−* SCNT (CD45.2+, Thy1.2+) CD8+ T cells into naïve CD45.1+ WT recipients 1 hr prior to CPS vaccination. (**B**) Representative FACS profile shows frequency of antigen specific polyclonal endogenous and monoclonal donor CD8+ T cells D7 post CPS vaccination. (**C**) CXCR3 expression on total *tgd057*-specific donor CD8+ T cells on D7 post vaccination (left) and expressed as MFI (right). (**D**) *tgd057*-specific donor CD8+ T cells were analyzed for CXCR3 expression by first gating for CD62L and KLRG1 cell surface expression, then studying expression of CXCR3 in F2 and F3 stages. (**E**) Frequency of F1–F4 stages of WT CD8+ T cells and *IL-12rβ2−/−* SCNT CD8+ T cells D7 post-CPS vaccination. (**F**) IFN-γ production by WT SCNT and *IL-12rβ2−/−* SCNT CD8+ T cells was analyzed after CPS restimulation. Data are from 3 independent experiments consisting of 4–8 mice per group per experiment. Mean ± SEM. Data were analyzed using unpaired t test, and (**D**) Bonferroni post-hoc test. (**E**) Mann–Whitney post-doc test; *p < 0.01, ***p < 0.001.

*tgd057*-specific CD8+ T cells (*Figure 6A*) without attenuating KLRG1 upregulation or IFN-γ production (*Figure 6B,C*).

The fact that blockade of IFN-γ can also prevent the downregulation of CXCR3 on KLRG1+ CD8+ T cells to the same extent as neutralization of IL-12 does, prompted us to ask if IFN-γ can suppress CXCR3 on CD8+ T cells similarly to IL-12. We exploited the fact that IL-12 deficient F3 stage KLRG1+ CD8+ T cells fail to decrease CXCR3 and used these cells as an in vitro model system to directly interrogate what factors can cause CXCR3 downregulation. Upon exposure and binding to its chemokine ligands, surface expressed CXCR3 is rapidly downregulated, endocytosed and degraded, but is quickly replenished by newly synthesized protein (*Meiser et al., 2008*). We therefore used CXCL10 exposure to 'strip' pre-existing CXCR3 from D7 CPS vaccinated IL-12p35 deficient splenocytes and determined whether IFN-γ or IL-12 can maintain the expression of CXCR3 on CD8+ T cells at

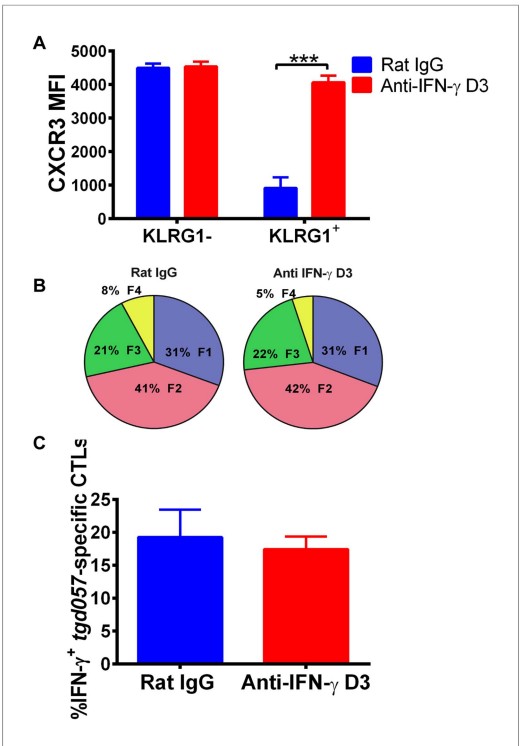

**Figure 6**. Neutralization of IFN-γ is sufficient to prevent CXCR3 downregulation on effector CD8⁺ T cell precursors. Anti-IFN-γ mAb was administered i.p. on D3 post-CPS vaccination. Mice were sacrificed on D5 post-CPS vaccination and *tgd057*-specific CD8⁺ T cells were analyzed for (**A**) CXCR3 expression, (**B**) CD62L and KLRG1 surface expression on *tgd057*-specific CD8⁺ T cells and (**C**) IFN-γ production post in vitro CPS restimulation in *tgd057*-specific CD8⁺ T cells. Data are from 2 experiments consisting of 3–5 mice per group per experiment. Mean ± SEM, paired t test and Holm-Sidak post-hoc test, ***p < 0.001.

a low level. As expected, the addition of IP-10 rapidly decreases the expression of CXCR3 on CD8⁺ T cells lacking IL-12 signals (*Figure 7A*, left), but importantly, the addition of IFN-γ and not the addition of CXCL10 alone, maintains the low levels of CXCR3 on these CD8⁺ T cells, to a similar extent as the addition of IL-12 (*Figure 7A*, right). This result suggests that IFN-γ may be acting directly on the developing CD8⁺ T cell to suppress CXCR3 expression downstream of IL-12. Consistent with this notion, addition of IFN-γ alone to highly purified CD8α⁺ TCRβ⁺ T cells from D7 CPS vaccinated IL-12p35 deficient spleens still maintains the low levels of CXCR3 expression (*Figure 7B*) without the need for IL-12-producing accessory cells, indicating that IFN-γ can act directly on the CD8⁺ T cell.

Overall, our data suggests there is late production of IL-12, which indirectly (putatively through an IFN-γ/CXCL9/10 mediated pathway) acts on the precursor effector CD8⁺ T cells to downregulate CXCR3 in the RP. To directly examine if there is late production of IL-12, we analyzed splenocytes from D3 CPS vaccinated mice after brefeldin A injection. CD11c⁺ CD11b⁺ DCs (myeloid DCs [mDCs]), but not CD8α⁺ DCs produced IL-12 (*Figure 7C*, D3). This result confirms that there is late production of IL-12 by a subset of mDCs distinct from the CD8α⁺ DCs known to be the primary producers of IL-12 during early *T. gondii* infection (*Figure 7C*, 4 hr). These mDCs may also be located in the RP where the precursor effector CD8⁺ T cells are trafficking to and subsequently downregulating CXCR3 (*Figure 3*). We also found that CD4⁺ and CD8⁺ T cells (*Figure 7D*), and to a much lesser extent NK cells, are producing IFN-γ in the spleen on day 3, suggesting that these cell types

are putatively responding to the IL-12 derived from these myeloid APCs. Finally, we find that the same myeloid APC subsets producing IL-12 are also the same cells producing CXCL9 and CXCL10 (*Figure 7C*) presumably in response to the IFN-γ derived from T_H1 CD4⁺ and CD8⁺ T cells. To determine whether the IL-12-producing mDCs are derived from inflammatory monocytes (*Nakano et al., 2009*), we vaccinated CCR2-GFP reporter mice (*Serbina et al., 2009*; *Espinosa et al., 2014*). Indeed, nearly all of the IL-12/chemokine -positive DCs on D3 are CCR2⁺ (*Figure 7—figure supplement 1A*). To evaluate the role of this late IL-12 producing APC population, we transiently depleted CCR2⁺ monocytes/derivatives on D2.5 by administering diptheria toxin to CCR2-DTR mice (*Hohl et al., 2009*). Transient ablation resulted in markedly diminished KLRG1⁺ (F3 and F4) subpopulations and attenuation in CXCR3 downregulation in the KLRG1⁺ CD8⁺ T cells. Although our data shows that CD8⁺ T cells are major IFN-γ producers at day 3, this response is probably not parasite antigen specific because, as indicated by the data above (*Figure 1A*), clonal bursting does not occur until after day 4. Taken together, our data suggests a three-cell model for how IL-12 indirectly leads to CXCR3 downregulation (*Figure 7E*). In the RP, bystander T_H1 CD4⁺ and CD8⁺ T cells are the cellular targets of the IL-12 produced by inflammatory monocyte-derived mDCs, which in response produce IFN-γ. IFN-γ then can directly target both mDCs to produce chemokines and the developing CD8⁺ T cells. The downregulation of CXCR3 on CD8⁺ T cells is driven by both chemokine-induced receptor downmodulation and by IFN-γ-mediated suppression of further CXCR3 expression.

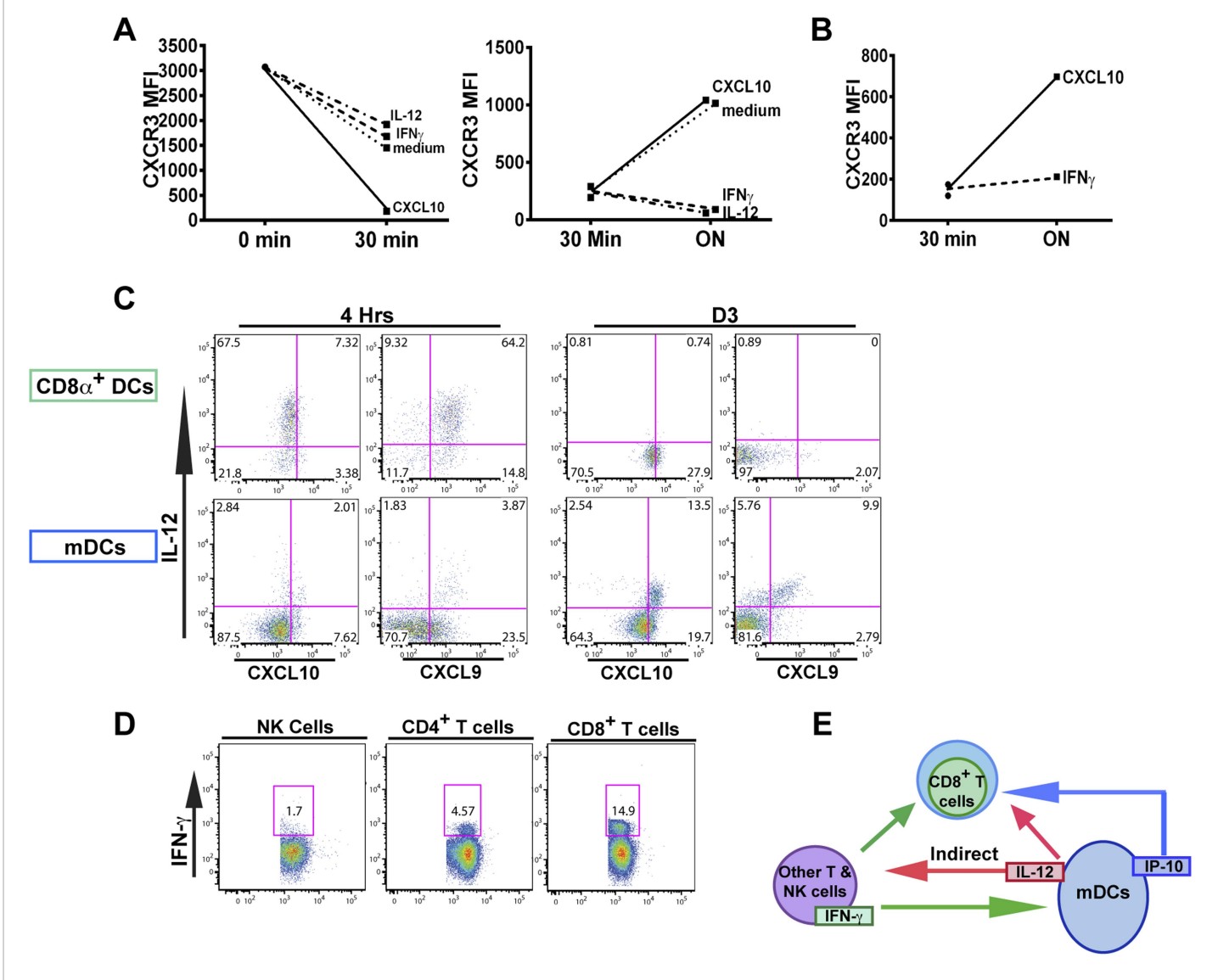

**Figure 7**. An alternative pathway involving IFN-γ and IFN-γ-inducible chemokines mediates suppression of CXCR3 on late-stage effector CD8+ T cell precursors. CXCL10, IL-12, or IFN-γ was added in vitro to (**A**) total splenocytes or (**B**) purified CD8+ T cells from D7 IL-12p35−/− mice post CPS vaccination. Cells were first 'stripped' of CXCR3 surface expression by addition of CXCL10 for 30 min (left panel), washed, and then incubated overnight (ON) with chemokine or cytokines. KLRG1+ CD8+ T cells were analyzed for CXCR3 expression. (**C**) Representative FACS profiles of ex vivo ICS of IL-12 and chemokines from 4 hr or D3 after CPS vaccination. Splenocytes were analyzed for IL-12p40, CXCL9 (MIG) and CXCL10 (IP-10) production in CD11c+ CD8α+ dendritic cells (DCs), and CD11c+ CD11b+ myeloid DCs (mDCs). (**D**) Representative FACS profiles of IFN-γ production from NK, CD4+ and CD8+ T cells in spleens of D3 CPS vaccinated WT mice. (**E**) Three cell model showing that CD8+ T cells are not the direct targets of IL-12. Rather IL-12 acts through bystander T cell production of IFN-γ, which targets both the developing CD8+ T cells and mDCs to secrete IFN-γ-inducible chemokines. CXCR3 chemokine ligands together with IFN-γ directly act on the developing CD8+ T cells to downregulate CXCR3 expression. (**A** and **B**) Data are from 4 experiments of 3–5 pooled IL-12p35−/− CPS vaccinated mice each, Mean ± SEM. (**C** and **D**) Data are from 2 experiments with 3–4 WT mice each.

The following figure supplement is available for figure 7:

**Figure supplement 1**. mDCs producing IL-12 and CXCL9 are CCR2+ and their depletion attenuates CD8+ T cell differentiation.

## Generation of KLRG1+ effector CD8+ T cells occurs after cessation of proliferation within the splenic RP

Taken together, our data suggest that most of the differentiative steps effector CD8+ T cells undergo occur extrafollicularly, raising the question of where their effector precursor proliferation takes place. A predominant view is that activation and proliferation of CD8+ T cells occurs at the outer zone of the T cell area bordering B cell follicles and subsequently exits via bridging channels to the splenic RP (*Khanna et al., 2007*). Therefore, we sought to determine whether CD8+ T cell proliferation occurs mainly in the white pulp and completes their differentiation in the RP as post-mitotic cells. We combined EdU labeling with intravascular staining during day 5 CPS-vaccination to localize where CD8+ T cell proliferation occurs in the context of the sequence of their differentiation. We found that a significantly greater proportion of proliferating CD8+ T cells (labeled within a narrow 2 hr period) were localized to the RP (*Figure 8A*, left) and comprised predominantly of CD62L−CD8+ T cells (F2 and F3) (*Figure 8A*, right). In fact, on a per cell basis, the proliferative rate

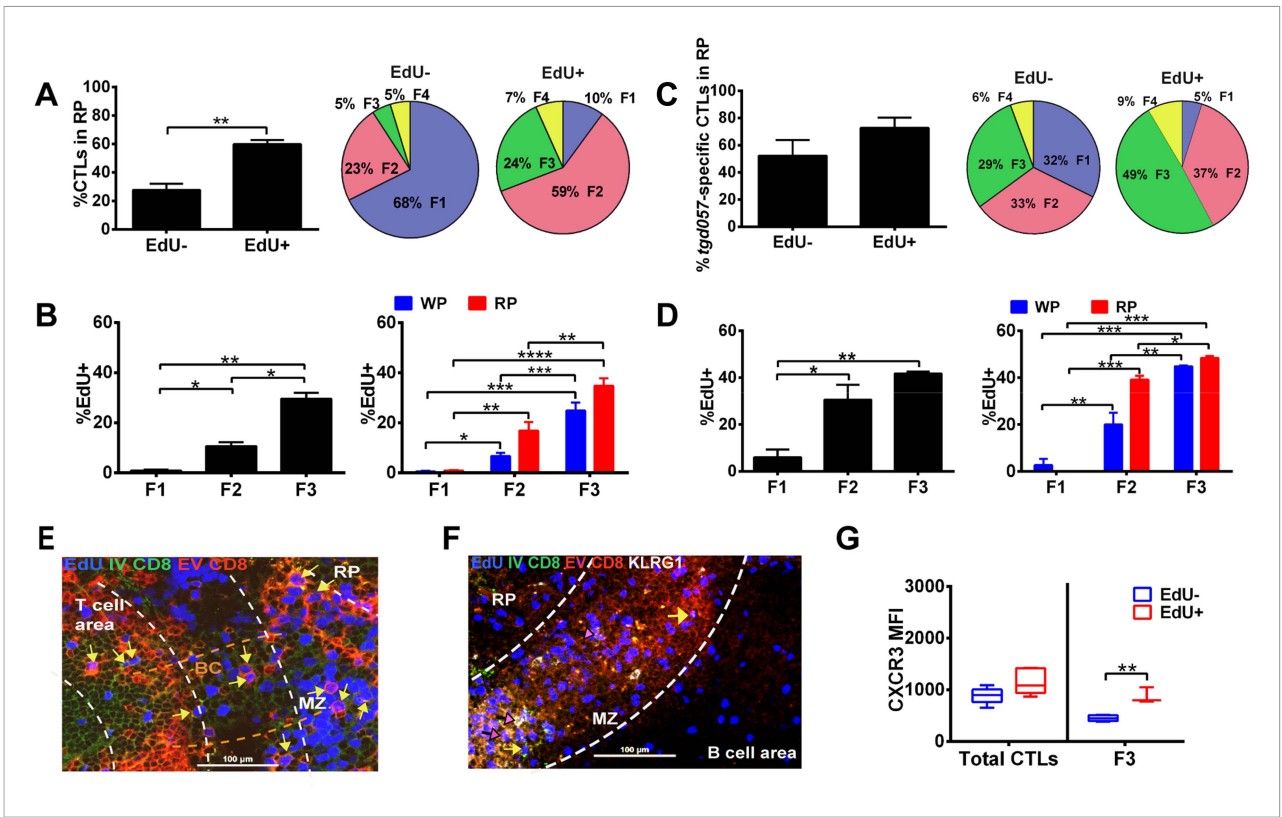

**Figure 8**. Generation of terminally differentiated CD8+ T cells occurs after cessation of proliferation in the splenic RP. D5.5 post CPS vaccinated WT mice were first injected i.p. with EdU 2 hr prior to sacrifice, then injected i.v. with 4 μg of anti-CD8-FITC antibody and sacrificed within 2–3 min. The differential localization and fractional distribution of F1–F4 stages in EdU− and EdU+ (**A**) for total CD8+ T cells and (**C**) *tgd057*-specific CD8+ T cells are shown. The rate of EdU-positivity as a function of differentiation stage (**B**, **D** left panels) or differentiation stage and i.v. anti-CD8 accessibility (**B**, **D** right panels) are shown for total CD8+ T cells (**B**) and *tgd057*-specific CD8+ T cells (**D**). Immunofluorescence images showing D5.5 splenic localization of EdU+ and double-positive for IV α-CD8 and ex vivo α-CD8 antibodies (**E**), and KLRG1 (**F**) staining. Splenic compartments are represented as follows: dashed line, MZ border; RP–red pulp; BC–bridging channel; MZ–marginal zone. Sections were stained for IV α-CD8 (green, CD8), ex vivo α-CD8 (red, EX CD8), EdU (blue), α-KLRG1 (white). Key to symbols: Yellow arrows: EdU+ IV CD8+ EV CD8+, Pink Arrows: EdU+ KLRG1+ IV CD8+ EV CD8+. Expression of CXCR3 in proliferating and non-proliferating CD8+ T cells (**G**). Data are from 2 experiments of 3–5 CPS vaccinated mice each. Immunofluorescence represents more than six sections per mouse taken from three mice. Mean ± SEM. Student's *t*-test or ANOVA analysis was done Holms-Sidak post hoc test, *p < 0.01, **p < 0.001 and ***p < 0.0001. Refer to *Figure 8—figure supplement 1* for IHC image of EdU+ CD8+ T cells in splenic bridging channels.

The following figure supplement is available for figure 8:

**Figure supplement 1**. CD8+ T cells in bridging channels are proliferating.

increases as cells progress from F1 to F3 stages (*Figure 8B*, left) and exhibit the same skewed extrafollicular localization of cycling CD62L⁻ CD8⁺ T cells (*Figure 8B*, right). A similar pattern of increased EdU labeling of CD62L⁻ fractions (F2 and F3) and a skewing towards RP localization of proliferative cells were evident in *tgd057*-specific CD8⁺ T cells (*Figure 8C,D*). These results indicate that, unlike the predominant view, CD8⁺ T cell proliferation is not restricted to the T cell area. To more precisely localize the extrafollicular sites of CD8⁺ T cell proliferation, we performed immunohistochemistry of D5.5 spleens immediately after IV injection of FITC-labeled anti-CD8α. As shown in *Figure 8E*, EdU-positive cells double-labeled with IV injected anti-CD8α and ex vivo stained anti-CD8α antibodies are readily observed in the MZ (marked with yellow arrows in *Figure 8E*) and in bridging channels (*Figure 8—figure supplement 1*). Extrafollicular proliferative CD8⁺ T cells are also observed in the RP, many of which are also KLRG1⁺ (marked with pink arrows in *Figure 8F*). Interestingly, these data also indicate that clonal proliferation does not cease when CD8⁺ T cells acquire the effector cell marker, KLRG1. This raises the question of whether CXCR3 downregulation occurs concomitantly or subsequent to CD8⁺ T cell proliferation in the KLRG1⁺ (F3) compartment. As *Figure 8G* indicates, CXCR3 downregulation is observed in non-proliferating KLRG1⁺ CD8⁺ T cells.

## Discussion

In this report, we identify the sequence of events (schematically depicted in *Figure 9*) that naïve CD8⁺ T cells undergo as they differentiate into effector CD8⁺ T cells in response to infection with the intracellular protozoan pathogen, *T. gondii*. We further delineate the steps where the pro-inflammatory cytokine, IL-12, is required for effector CD8⁺ T cell differentiation. Upon activation, CD44, a surface receptor necessary for T cell adhesion to extracellular matrix proteoglycans, is rapidly upregulated and then L-selectin (CD62L) is downregulated in an IL-12-independent manner (*Figure 1*). CXCR3 is expressed early in the CD44^Hi, CD62L⁺ (F1) CD8⁺ T cell stage, and remains highly expressed through the CD44^Hi, CD62L⁻ (F2) CD8⁺ T cell stage (*Figure 2*). This early expression of CXCR3 is T-bet dependent but IL-12 independent (*Figure 2*). While still in the white pulp and even prior to CD8⁺ T cell expansion (*Figures 1, 3*), a few CD44^Hi, CD62L⁻ CD8⁺ T cells begin to acquire KLRG1 surface expression in an IL-12 dependent manner. As CD8⁺ T cell numbers increase and outmigrate to the RP, KLRG1 expression escalates (*Figure 3*). As shown in *Figure 8* we are able to reveal, for the first time, that CD8⁺ T cell proliferation does not only occur exclusively in the white pulp (*Figure 8A–D*). Unexpectedly, we found that most of the proliferating CD8⁺ T cells are CD62L⁻ (F2 and F3 stage CD8⁺ T cells) and many of these cycling effector precursors are observed to be outmigrating into the MZ (*Figure 8*) via bridging channels (*Figure 8—figure supplement 1*) (*Khanna et al., 2007*). Finally, these extrafollicular KLRG1⁺ precursor effector CD8⁺ T cells downregulate CXCR3 expression in an IL-12 dependent fashion. CXCR3 downregulation by IL-12 is indirectly mediated by IFN-γ and IFN-γ-dependent chemokines, which are produced by other T lymphocytes and mDCs, respectively (*Figure 7*), in the MZ where the extrafollicular CD8⁺ T cells have been shown to cluster and interact with inflammatory myeloid cells producing IL-12 in a CXCR3 dependent manner (*Kurachi et al., 2011*). Our report outlines a detailed order of events during the differentiation of naïve CD8⁺ T cells to KLRG1⁺ CXCR3⁻ effector CD8⁺ T cells and, together with several recent reports (*Khanna et al., 2007*; *Kohlmeier et al., 2011*; *Kurachi et al., 2011*; *Groom et al., 2012*; *Sung et al., 2012*), highlights the importance of the MZ and RP as extrafollicular sites for their proliferation and differentiation.

Our in-depth analysis of IL-12 regulated checkpoints during the early effector CD8⁺ T cell programming in response to *T. gondii* vaccination has provided several new insights. While IL-12 has been thought to be the major signal 3 cytokine needed for in vitro proliferation (*Curtsinger et al., 1999*; *Valenzuela et al., 2002*) and augmentation of the immune synapse (*Markiewicz et al., 2009*), we found that IL-12 does not provide an obligatory signal for *T. gondii*-specific CD8⁺ T cell clonal expansion in vivo, but rather is critical for the early KLRG1 expression on precursor effector CD8⁺ T cells. Although KLRG1, IFN-γ and CXCR3 are under transcriptional control of T-bet, the expression pattern of CXCR3 is opposite to the expression patterns of KLRG1 and IFN-γ, such that CXCR3 expression is increased early during CD62L⁺ F1 and CD62L⁻ F2 stages of CD8⁺ T cell differentiation, while KLRG1 is expressed during the later stages of effector differentiation (*Figure 2*). While CXCR3 is expressed early and its upregulation is IL-12 independent, we show that that once precursor effector CD8⁺ T cells upregulate KLRG1, CXCR3 expression is downregulated in an IL-12 dependent manner (*Figures 2, 4*). Even though IL-12 is required for both the upregulation of KLRG1 and the downregulation of CXCR3,

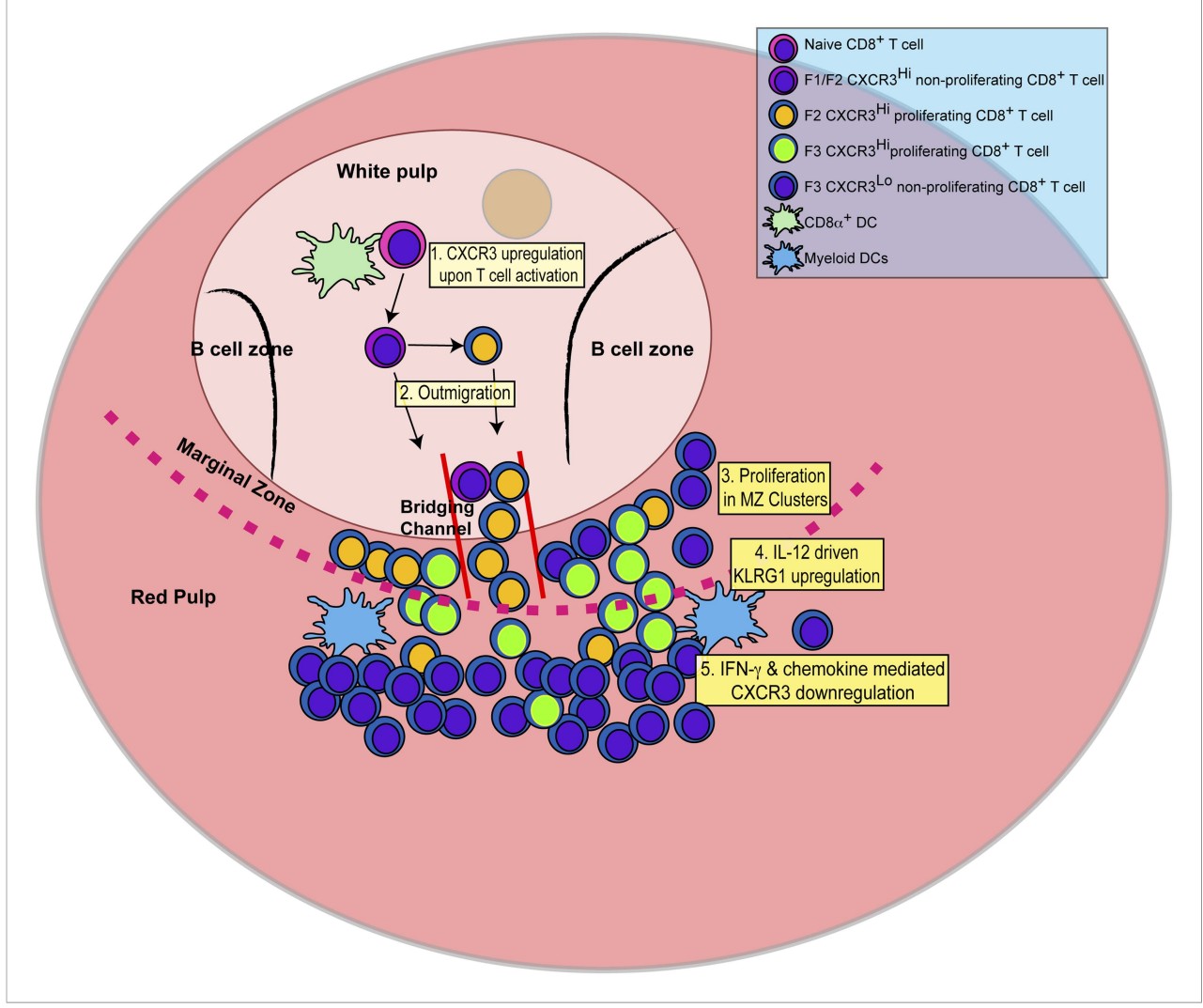

**Figure 9**. Schematic depicting the sequence of events leading to the generation of late-stage KLRG1[+] effector CD8[+] T cells. 1: While in the white pulp, CD8α[+] DCs activate CD8[+] T cells causing the upregulation of CXCR3 expression. 2: CXCR3[Hi] CD62L[Lo] CD8[+] T cells outmigrate towards RP and begin to proliferate. 3: Upon outmigration through bridging channels, CXCR3[Hi] CD62L[Lo] CD8[+] T cells continue to proliferate in clusters found in the MZ. 4: Exposure to IL-12 produced by mDCs in the MZ/RP areas induces KLRG1 upregulation. 5: Subsequent exposure to IFN-γ and IFN-γ-inducible chemokines downregulate CXCR3 expression on RP-localized KLRG1[+] effector CD8[+] T cells.

these two effects can be temporally and spatially dissociated (*Figures 3, 4*). Furthermore, while the IL-12 effects on KLRG1 and IFN-γ are acting on the CD8[+] T cells themselves, we found, surprisingly, that downregulation of CXCR3 is regulated by IL-12 in a non-CD8[+] T cell autonomous fashion (*Figure 5*). This latter finding seems to be at odds with a previous report that IL-12 directly regulates the suppression of CXCR3 expression of effector CD8[+] T cells (*Slutter et al., 2013*). In this previous study, *Slutter et al. (2013)* also used a chimeric approach with IL-12rβ1 deficient and WT OT-1 cells to demonstrate that IL-12 receptor signaling has negative regulatory effects on CXCR3 expression during influenza priming and that this control is CD8[+] T cell-intrinsic. Our results indicate a similar conclusion when we analyzed CD8[+] T cells without regard for KLRG1 expression (*Figure 5*). However, when we analyze CXCR3 expression on KLRG1[−] vs KLRG1[+] CD8[+] T cells expression, we arrive at a different conclusion (*Figure 5*). In the absence of IL-12 receptor signaling, we show that KLRG1[+] CD8[+] T cells maintain low CXCR3 expression (*Figure 5*). The appearance of CD8[+] T cell autonomous control can be explained by population skewing at the expense of KLRG1[+] CD8[+] T cells, which is the only stage where CXCR3 is decreased. Thus, even in the influenza system, IL-12 probably indirectly downregulates CXCR3.

The data from both our chimeric transfer and late in vivo IFN-γ neutralization experiments indicate that IL-12 does not directly regulate CXCR3 downmodulation during precursor effector CD8+ T cell development (*Figures 6, 7*), but instead acts through an indirect pathway by which IFN-γ suppresses CXCR3 expression. In support of this model, we find that there is production of IFN-γ by bystander $T_H1$ T cells and that IFN-γ maintains CXCR3 suppression in vitro. Thus, we propose a three-cell model (*Figure 7*) for CXCR3 downregulation, in which the developing effector CD8+ T cell is the direct target of IFN-γ derived from bystander $T_H1$ T cells and IFN-γ-induced chemokines produced by mDCs. In this model, IL-12 acts by inducing IFN-γ production from bystander $T_H1$ T cells rather than the developing effector CD8+ T cells themselves. The mechanism of CXCR3 suppression by IFN-γ is currently not clear, but it will be interesting to find out whether it uses the same ID2-mediated pathway described for IL-12 (*Yang et al., 2011*; *Knell et al., 2013*). It is also unclear what signals drive the production of IL-12 by mDCs at later time points after the initial IL-12 production by CD8α+ DCs has waned. It is possible that residual non-replicating *T. gondii* parasites persist (*Dupont et al., 2014*) and, in addition, IFN-γ itself acting together with accessory signals from bystander lymphocytes provides inductive signals for mDC production of IL-12 (*Ariotti et al., 2014*; *Slutter and Harty, 2014*). Regardless of these uncertainties, our identification of IFN-γ as the critical mediator for CXCR3 downregulation provides a common mechanism for how multiple proinflammatory cytokines (*Yang et al., 2011*) can redundantly downmodulate CXCR3 on developing effector CD8+ T cells.

Our report highlights that the expression pattern of CXCR3 is highly complex as CD8+ T cells progress through distinct stages of differentiation. Although CXCR3 expression has been classically associated with IL-12 driven type-1 effector adaptive immune responses, we show that CXCR3 is expressed very early in the CD62L+ KLRG1− F1 stage and thus allowing CD62L− KLRG1− F2 stage effector precursor CD8+ T cells to outmigrate to the MZ and the RP, where they are further exposed to proinflammatory signals and differentiate into effector CD8+ T cells. The same CXCR3 mediated outmigration has been reported for primary CD4+, CD8+ T cells and central memory T cells (*Kurachi et al., 2011*; *Groom et al., 2012*; *Sung et al., 2012*; *Kastenmuller et al., 2013*), arguing for the existence of a common extrafollicular pathway for type-1 effector cell generation. Following proinflammatory cytokine and chemokine exposure KLRG1+ CXCR3Hi precursor effector CD8+ T cells gain the ability to produce high levels of IFN-γ while downregulating CXCR3 expression. The observation that outmigrating differentiated effector CD8+ T cells found in the splenic RP are CXCR3Lo while published studies have clearly shown that tissue associated effector CD8+ T cells are mostly CXCR3Hi (*Kohlmeier et al., 2011*; *Harris et al., 2012*; *Slutter et al., 2013*; *Ochiai et al., 2015*) raises the question of whether they reacquire CXCR3 once they reach infected tissue sites. A recent study of Mtb-infected lung tissues has demonstrated the existence of two distinct subpopulations of CD4+ $T_H1$ cells that differ in their accessibility to an intravenous fluorescent antibody tracer (*Sakai et al., 2014*). The tracer-accessible subpopulation resembled our F3 stage effector CD8+ T cells being KLRG1+ CXCR3Lo CX3CR1Hi and was IFN-γHi. In contrast, the other subpopulation more deeply localized in the tissue parenchyma were CXCR3Hi and exhibited a highly activated phenotype with expression of CD69 and PD-1, but lower in KLRG1 expression and IFN-γ production. Therefore, one interesting scenario is by virtue of their CX3CR1 expression, the F3 stage effector cell circulates in the blood and serves a patrolling function similar to CX3CR1Hi monocytes (*Auffray et al., 2007*). They could then regain CXCR3 expression and function as effector CD8+ T cells in parasite-infected tissues. Future studies should investigate the reciprocal patterns of expression of CXCR3 and CX3CR1 and their counter regulation.

Using immunohistochemistry methods *Khanna et al. (2007)* demonstrated the outmigration of antigen-specific CD8+ T cells through 'bridging channels' into the MZ and splenic RP. *Kurachi et al. (2011)* further reported that CXCR3 enabled CD8+ T cells to cluster with IL-12 producing CD11c+ DCs in the MZ and RP. Now, we show that precursor effector CD8+ T cells in the bridging channels and MZ are, in fact, actively proliferating and that they receive IL-12 signals to complete their differentiation in the RP. Collectively, these findings highlight a previously unrecognized extrafollicular pathway for the generation and maturation of primary effector CD8+ T cells. Interestingly, in the B cell response, there is an analogous and well-recognized extrafollicular differentiation pathway of IgG class switched plasma cells producing the primary antibody response to T cell-dependent antigens (*MacLennan et al., 2003*; *Kurosaki et al., 2015*). By analogy to the way that the non-germinal center plasma cell response is favored by strong B cell receptor signaling and antigen presentation by MZ localized DCs (*Paus et al., 2006*; *Phan et al., 2006*; *Chappell et al., 2012*), the extrafollicular generation of effector

CD8[+] T cells is also likely regulated by differential T cell receptor signaling (*Bernhard et al., 2015*) and local cytokine and chemokine milieu created by specialized pathogen-engaged APCs interacting with bystander lymphocytes in the MZ and RP. The exposure of effector precursors to this extrafollicular tissue niche could provide major extrinsic factors that determine the observed interclonal and, perhaps, intraclonal variability in effector lymphocyte fates that is independent of their intrinsic TCR-peptide:MHC affinity and dwell time (*Plumlee et al., 2013*; *Tubo et al., 2013*).

## Materials and methods

### Mice

WT C57BL/6J, *IL-12rβ2*[−/−] (B6.129S1-*Il12rb2*[tm1Jm]/J) male, CD45.1[+] recipients (B6.SJL-*Ptprc*[a]*Pep*[c]/BoyJ) and *T-bet*[−/−] (B6.129S6-*Tbx21*[tm1Glm]/J) mice were purchased from The Jackson Laboratory (Bar Harbor, ME). *IL12p35*[−/−] mice (B6.129S1-*Il12a*[tm1Jm]/J) were bred in house. SCNT mice cloned from a single CD8[+] T cell specifically expressing TCRαβ for *tgd057*/SVLAFRRL antigen were a generous gift from Hidde L Ploegh at MIT (*Kirak et al., 2010b*). To generate *Thy1.1*[+] *tgd057-specific SCNT mice*, SCNT females were crossed with a Thy1.1[+] male. The F1 progeny were backcrossed to SCNT females and F2 progeny expressing Thy1.1 and TCRαβ specific for *tgd057*/SVLAFRRL antigen were selected. To generate *IL-12rβ2*[−/−] *tgd057-specific SCNT mice*, SCNT females were crossed with *IL-12rβ2*[−/−] males. The F1 progeny with tetramer-binding CD8[+] T cells were intercrossed. The F2 progeny were then genotyped to select for *IL-12rβ2*[−/−] using tail or ear tissue and phenotyped for inheritance of TCRαβ specific for *tgd057*/SVLAFRRL antigen. CCR2-DTR depleter (CCR2-DTR) and CCR2-reporter (CCR2-GFP) strains previously generated on a C57BL/6 background as previously described were used to investigate the role of mDCs during CD8[+] T cell differentiation (*Hohl et al., 2009*; *Serbina et al., 2009*; *Espinosa et al., 2014*). Breeding, handling and housing of all mice was under specific pathogen-free conditions at Rutgers University (formerly University of Medicine and Dentistry of New Jersey) in Newark, NJ and according to the Rutgers Institutional Animal Care and Use Committee guidelines. All mice used for experiments were age- and sex-matched.

### Vaccination and parasites

Monolayers of human foreskin fibroblasts (HFF) were infected with *T. gondii* tachyzoites that have a disruption in the carbomyl phosphate synthetase (*cps 1-1* or CPS) II gene, cultured in DMEM (Invitrogen, Carlsbad, CA) supplemented with uracil (*Fox and Bzik, 2002*). Fresh tachzyoites were isolated from approximately 80% lysed HFF cultures and irradiated with 150gy [137]Cs before injection.

### Tissue preparations and in vitro IFN-γ restimulation

Mice were sacrificed on specified days post CPS vaccination. Spleens of infected and uninfected mice were isolated by physical disruption and passage through 70 μm nylon mesh (BD Falcon), washed and erythrocytes lysed using a hypotonic Tris-buffered NH4Cl solution. PECs were isolated by peritoneal lavage with RPMI 1640 (Invitrogen) supplemented with 2% FBS, 1% P/S, 50 μM β-mercatopethanol. Live cells from spleens and PECs were counted using trypan blue dye exclusion.

To restimulate cells in vitro, $4{-}6 \times 10^{6}$ cells from spleens and PECs of infected and uninfected animals were plated and restimulated with live CPS tachyzoites (MOI of 0.1 or MOI of 0.3) and incubated for 10 hr at 37°C. GolgiStop (BD Bioscience, San Jose, CA) was added 4 hr prior to end of in vitro incubation.

### Tetramer-based enrichment assay

Splenocytes were stained with PE-streptavidin-biotin SVLAFRRL:H2-K[b] tetramers for 1 hr on ice in the dark, washed with FACS buffer, then incubated with anti-PE magnetic microbeads (Miltenyi Biotec, Auburn, CA) according to manufacturer's instructions. Mix was then applied to LS magnetic column (Miltenyi Biotec), and positively selected cells were collected and analyzed by flow cytometry.

### Flow cytometry

Cell surface antibody and tetramer staining was performed concurrently in FACS buffer for 1 hr on ice in the dark, washed and fixed. To stain for intracellular cytokines, samples from in vitro restimulated

cultures were permeabilize and stained with cytokine antibodies. Samples were washed and resuspended and FACS analyzed. Flow cytometry data were acquired on BD LSRII and analyzed with FlowJo software (Tree Star, Ashland, OR).

Mouse specific antibodies were purchased from eBioscience (San Diego, CA): CD44-PE Cy7, IFN-γ-eFluor450, CD8α-Alexa Fluor 700, TCRβ-PerCP-Cy5.5, KLRG1-APC, Thy1.1-eFluor 450 or from BD Bioscience: CD62L-APC-eFluor780 and CXCR3-FITC. PE-streptavidin-biotin SVLAFRRL: H-2K$^b$ tetramers were prepared as previously described (*Wilson et al., 2010*).

## Adoptive transfer

CD8$^+$ T cells were negatively selected from naïve Thy1.1$^+$ *tgd057*-specific SCNT mice, and *IL12rβ2$^{-/-}$ tgd057*-specific SCNT mice. Negative selection was performed using magnetic micro beads (Miltenyi Biotec). A small aliquot of cells were surface stained for CD8α, TCRβ, and tetramer, fixed and analyzed for purity. The frequency of CD8α$^+$ *tgd057*-specific cells was used to normalize the numbers of *tgd057*-specific CD8$^+$ T cells from the negatively selected samples. For adoptive transfer studies, Thy1.1$^+$ *tgd057*-specific SCNT CD8$^+$ T cells were resuspended in PBS and intravenously (i.v.) injected into either naïve WT or *IL-12p35$^{-/-}$* recipients. For adoptive co-transfer experiments, a 1:1 mix of 500 donor Thy1.1$^+$ *tgd057*-specific SCNT CD8$^+$ T cells and 500 donor *IL12rβ2$^{-/-}$ tgd057*-specific SCNT CD8$^+$ T cells were mixed and i.v. injected into naïve WT CD45.1$^+$ congenic recipient mice.

## Intravascular staining to discriminate CD8$^+$ T cells in splenic white and RP regions using fluorescently labeled anti-CD8α antibody

D4, D5, D5.5 or D7 CPS vaccinated mice were i.v. injected with 2.5–4 μg of fluorescently labeled anti-CD8α antibody. The mice were sacrificed 2–3 min post injection, and splenocyte single-cells suspensions were prepared and stained for CD8α in a different fluorochrome along with other surface antibodies for flow cytometry analysis. Cells in contact with blood circulation are labeled by intravenously injected anti-CD8$^+$ antibody, while cells in the splenic white pulp are not.

## In vivo EdU for CD8$^+$ T cell proliferation

D4 or D5 CPS vaccinated mice were i.p. injected with EdU labeling reagent (10 ml/kg body weight, Invitrogen, C-10418) dissolved in sterile PBS 2 hr prior to sacrifice. EdU detection was performed according to manufacturer's manual using Click-iT EdU- Pacific Blue labeling kit after splenocytes were surface stained and fixed.

## Immunohistochemistry

D4.5 and D5.5 CPS vaccinated spleen were snap frozen in OCT compound (Tissue-Tek, Sakura Finetek, Torrance, CA). 6–8 μm sections were sectioned and briefly rehydrated in sterile PBS, surface stained for anti-CD8α and anti-KLRG1, fixed with 4%PFA then EdU detection was performed according to manufacturer's manual (Invitrogen, C-10418) using Click-iT-EdU Pacific Blue labeling kit. Sections were mounted with Prolong Gold Antifade Mountant (Invitrogen, P10144). Immunofluorescent images were captured on a Nikon A1R Si confocal microscope equipped with a 20× Plan Apo VC dry objective and analyzed using NIS Elements C3.13.01 software (Nikon, Melville, NY, United States).

## Neutralization of endogenous IL-12p40 or IFN-γ in vivo

On day 0, day 3 or both days, each mouse was treated via intraperitoneal (i.p.) injection with 200 μg of anti-IL-12p40 monoclonal antibody (C17.8), anti-IFN-γ (XMG.1) or the isotype control, 2A3 clone. 4 hr post antibody injection, on day 0, mice were vaccinated with 2.5 × 10$^6$ irradiated CPS parasites. Spleen and PECs were harvested for analysis 5 days post CPS vaccination.

## CCR2$^+$ monocyte depletion

CCR2$^+$ monocytes were depleted in CCR2-DTR mice and control CCR2-DTR negative littermates by i.p. administration of 250 ng of diphtheria toxin 2.5 days post CPS vaccination (*Espinosa et al., 2014*). Diphtheria Toxin was purchased from List Biological Laboratories (Campbell, CA), and reconstituted at 1 mg/ml in PBS. Aliquots were stored in 80°C. Spleens were harvested on D5 post CPS vaccination, and *tgd057*-specific CD8$^+$ T cells were analyzed.

## Statistical analysis

Data are represented as Mean ± SEM. All error bars represent SEM. p values were calculated with Student's t-tests or ANOVA (GraphPad Prism) as indicated. p values of less than 0.05 were considered significant.

## Acknowledgements

We thank Hidde L Ploegh (Massachusetts Institute of Technology, Boston, MA) for generously providing our laboratory with tgd057-specific SCNT mice. We would also like to thank our colleagues, Andrew Marple and Yanlin Zhao for useful discussion regarding this work. This work is supported by NIH grant RO1 AI083405 awarded to GSY.

## Additional information

### Funding

| Funder | Grant reference | Author |
| --- | --- | --- |
| National Institute of Allergy and Infectious Diseases (NIAID) | RO1 AI083405 | George S Yap |

The funder had no role in study design, data collection and interpretation, or the decision to submit the work for publication.

### Author contributions

SS, Conception and design, Acquisition of data, Analysis and interpretation of data, Drafting or revising the article; GMG, AR, Drafting or revising the article, Contributed unpublished essential data or reagents; GSY, Conception and design, Analysis and interpretation of data, Drafting or revising the article

### Ethics

Animal experimentation: This study was performed in strict accordance with the recommendations in the Guide for the Care and Use of Laboratory Animals of the National Institutes of Health. All of the animals were handled according to approved institutional animal care and use committee (IACUC) of the Rutgers Biomedical Health Sciences, protocol number 13110. The protocol was approved by the Committee on the Ethics of Animal Experiments of Rutgers Biomedical Health Sciences.

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
