## [Decision Letter]

Thank you for submitting your work entitled “An extrafollicular pathway for the generation of effector CD8^+^ T cells driven by the proinflammatory cytokine, IL-12” for peer review at *eLife.* Your submission has been favorably evaluated by Tadatsugu Taniguchi (Senior Editor) and three reviewers, one of whom, Urszula Krzych, is a member of our Board of Reviewing Editors.

The reviewers have discussed the reviews with one another and the Reviewing Editor has drafted this decision to help you prepare a revised submission.

1) This is an interesting and important study that probes the sequence of events occurring during early CD8 T cell differentiation after vaccination with Toxoplasma gondii. Specifically, using tetramers and parasite-specific TCR transgenic T cells, the authors examine the regulation of KLRG1 and CXCR3 on effector CD8^+^ T cell populations and the role of IL-12, an inflammatory cytokine, that appears to direct the differentiation process intrinsically and extrinsically in regulating KLRG1/IFN-γ and CXCR3 expression on maturing CD8^+^ T cells. One of the highlights of the study is the extra-follicular pathway for the generation and maturation of effector CD8^+^ T cells. This is a well thought out and experimentally well executed manuscript, which addresses a timely matter in CD8^+^ T cell biology.

2) Although the reviewers find the results quite interesting, they also indicate that the study lacks a certain degree of rigor that weakens the conclusions. A few key experiments as proposed in the reviews from each individual reviewer might significantly enhance the quality of this manuscript and strengthen the author's conclusions. Specifically, the authors need to address the issue of IL-12 production and CXCL0/10 expression by myeloid DCs. Importantly, if the authors plan to resubmit the manuscript, it is strongly suggested that the hypothesis are stated with clarity, that the Results section is organized and written without ambiguity and that the Discussion is less repetitions of the Results. A point worth considering would be to streamline the whole manuscript and in particular the Results section. Comments from each reviewer are included as a guide for an acceptable resubmission of the manuscript.

Reviewer #1:

The authors describe their investigation of the relationship between IL-12 inflammatory cytokine and early differentiation stages of CD8^+^ T cells; one of the highlights of the study is the extra-follicular pathway for the generation and maturation of effector CD8^+^ T cells. I must say that I found the Results section of this paper rather challenging to read and thus it was difficult to understand the reasons for performing certain experiments. Simply said, I got lost in their KLRG1 cum CXCR3 cum IL-12 story.

One of the key results that prompted them to start this investigation was the observation that Tg T cell appear to have attenuated migration from the spleen to the peritoneum in IL-12p35-/- mice. This phenomenon appears only on about day 6 after infection, the height of adoptive immune response. They decided then to investigate if the expression of CXCR3 might be related to the attenuated migration (Figure 1). In my view, the reduced numbers of the Tgd057 cells could be attributed to many other factors, e.g. cell death in this mouse knockout.

The authors then assert that while the early expression of CXCR3 is IL-12 independent, it is T-bet dependent. Yet, in my reading of the data in 2B, it seems that WT KLRG1+ cells have the same profiles of CXCR3 as KLRG1+ cells in T-bet -/- mice. In addition, the reference to early vs. late events, is a bit confusing when the analysis that is performed reflects only day 7 post infection. Wouldn't it be more precise to address the different stage of CD8^+^ T cell differentiation instead?

Although the observations reported here depart from the accepted notion about CXCR3 expression on effector CD8^+^ T cells, these results may be significant for responses generated by protozoan parasites. Therefore, it may be worth asking for resubmission of this manuscript but with that clearly stated hypothesis, logically arranged and clearly written Results section and a Discussion that is much less repetitious of the Results. A point worth considering would be to streamline the Results section. For example, the first two paragraphs in the Results section could be expressed succinctly in one paragraph. In short, I find this to be a very complex study and it shouldn't be. I believe that the unduly complex nature stems from the disorganization of the Results section, which obviates the clarity of the intended story.

Reviewer #2:

This is an interesting and important study that probes the sequence of events occurring during CD8^+^ T cell priming and differentiation after vaccination with *Toxoplasma gondii*. The authors focus on antigen-specific CD8^+^ T cells through use of tetramers and parasite-specific TCR transgenic T cells, based upon previous studies by this group that identified an immunodominant CD8^+^ epitope. They use a really nice method of transferring a defined number of marked antigen-specific “calibrator” cells to accurately quantify what is happening to the endogenous population of parasite-specific CD8^+^ cells. These results are highly interesting, and they get to a level of depth that few or no other studies reach. However, there is a certain lack of rigor that weakens the conclusions and the events presented in the schematic figure (Figure 9).

A major point of the paper is that IL-12 mediated CXCR3 downregulation can be dissociated temporally from its effects on KLRG1 upregulation. However, this doesn't really come through in the data presented. A kinetic study of CXCR3 expression on the F1-F4 CD8^+^ cells (as they do for KLRG1 in Figure 1) would be helpful.

Another important focus of the paper is defining the CD8 differentiative events occurring in the red pulp. Whether a given cell is present or not in the red pulp is based largely upon whether the cell binds anti-CD8 Ab after brief in vivo pulse. The immunohistochemistry shown in Figure 8 which would validate these results is unclear and needs improvement.

In the authors' model, they speculate that early IL-12 is provided by CD8α^+^ DC, and later on in the red pulp region by other myeloid cell types. This would be important to establish. Similarly, they suggest several candidate cell types that could supply IFN-γ for IL-12 independent cell autonomous effects based upon observational data, without actually identifying the functionally important cells.

Reviewer #3:

In this manuscript Shah et al. examine the regulation of KLRG1 and CXCR3 on effector CD8^+^ T cell populations following T. gondii vaccination. Interestingly, they find CD8^+^ T cell intrinsic and extrinsic roles for IL-12 in regulating KLRG1/IFN-γ and CXCR3 expression, respectively. Moreover, they correlate the activity of IL-12 to the red pulp area of the spleen largely and this was also the location they observed the most cell proliferation (as determined by EdU incorporation). This is a well thought out and experimentally well executed manuscript, which addresses a timely matter in CD8^+^ T cell biology. A few key experiments might significantly enhance the quality of this manuscript and strengthen the author's conclusions.

1) Figure 1: After the tetramer-enrichment, especially at the early time-points (prior to day 3), have the authors included other activation markers such as CD44, CD11a, CD25, and/or CD69 to ensure that the F1 population (CD62L+ KLRG1-) is truly a ‘Tcm’ population and not the naive CD8^+^ T cell population, which would have the same CD62L and KLRG1 expression pattern?

2) Figure 2: The expression of CXCR3 early by activated CD8^+^ T cell being IL12-independent but Tbx21(T-bet)-dependent is very interesting? Have the authors looked at other factors that drive T-bet, such as IL-27? This would be very interesting to examine. But definitely should be discussed more thoroughly in the manuscript.

3) Figure 7: In vivo do IFN-γ receptor deficient CD8^+^ T cells have altered CXCR3 regulation like is suggested by the in vitro experiments present in this figure. Demonstrate this in vivo would be nice.

4) The finding that myeloid DCs express both IL-12 and CXCL9/10 at D3 post-challenge is very interesting. Can the author directly evaluate the importance of this cell population in regulating CXCR3 expression in vivo?

---

## [Author Response]

Reviewer #1:

*[…] In my view, the reduced numbers of the Tgd057 cells could be attributed to many other factors, e.g. cell death in this mouse knockout*.

We have also considered this possibility and have directly assessed apoptosis rates and found no differences between wild-type and knockout mice.

*A point worth considering would be to streamline the Results section. For example, the first two paragraphs in the Results section could be expressed succinctly in one paragraph*.

We have combined the first two paragraphs as suggested by Reviewer 1.

Reviewer #2:

*[…] A kinetic study of CXCR3 expression on the F1-F4 CD8 cells (as they do for KLRG1 in*
Figure 1*) would be helpful.*

We have not performed a detailed day-by-day analysis of CXCR expression as the reviewer suggested. Nevertheless, from the data we have presented in Figure 3, it is evident that KLRG1+ cells still express CXCR3 on day4 but by day 7 have downregulated chemokine receptor expression. In other experiments where the endpoint is conducted on day 5 (for example, Figure 4), the expression of CXCR3 is already depressed.

*Another important focus of the paper is defining the CD8 differentiative events occurring in the red pulp. Whether a given cell is present or not in the red pulp is based largely upon whether the cell binds anti-CD8 Ab after brief in vivo pulse. The immunohistochemistry shown in*
Figure 8
*which would validate these results is unclear and needs improvement*.

The in vivo anti-CD8 pulse labelling technique we used provided a highly quantitative and reproducible method for discriminating the localization of differentiation events we have outlined in this study. Because of these advantages, the usage and acceptance for this technique is on the rise, as reflected by the recent review in Nature Protocols (9:209, 2014) by David Masopust, Dan Barber and colleagues. The immunohistochemistry images we provided in Figure 8 and Figure 8—figure supplement 1 simply confirm that extrafollicular proliferation of CD8^+^ T cells does in fact occur, and further provide anatomical detail on in which subregions of the spleen these proliferative events occur. To improve the visualization, we increased the image brightness while decreasing contrast. An unwanted effect is the increase in “background” iv anti-CD8 staining in the white pulp, which we can readily distinguish from the true signal emanating from anti-CD8 staining of extrafollicular T cells because the distinctive punctate staining pattern on the cell surface. This punctate pattern may arise from capping during the intervening time between injection, sacrifice and processing for immunohistochemistry. (Should this reviewer desire, we can provide a panel showing the individual channels for the IHC staining we used to produce the merged image we show in Figure 8.)

It is important to note that our manuscript is the first direct evidence of T cell proliferation occurring outside of the white pulp. Mostly immunologists have been taught and think that T cell activation and proliferation occur within the periarteriolar region. Clearly, additional efforts have to made in future studies to further document and understand this phenomenon.

*In the authors' model, they speculate that early IL-12 is provided by CD8a DC, and later on in the red pulp region by other myeloid cell types. This would be important to establish. Similarly, they suggest several candidate cell types that could supply IFN-g for IL-12 independent cell autonomous effects based upon observational data, without actually identifying the functionally important cells*.

In response to this, we have further characterized the IL-12 producing myeloid DCs as being CCR2+, suggesting inflammatory monocytes are their origin. We further transiently depleted CCR2+ cells and show that such treatment recapitulates many facets of late IL-12 depletion, including loss of KLRG1+ cells and an attenuation in the down-regulation of CXCR3 expression. These new data are presented as Figure 8—figure supplement 1.

Reviewer #3:

*1)*
Figure 1*: After the tetramer-enrichment, especially at the early time-points (prior to day 3), have the authors included other activation markers such as CD44, CD11a, CD25, and/or CD69 to ensure that the F1 population (CD62L+ KLRG1-) is truly a ‘Tcm’ population and not the naive CD8*^*+*^
*T cell population, which would have the same CD62L and KLRG1 expression pattern?*

We have stained for CD44 but not CD11a, CD25 or CD69. We have provided the information regarding CD44 staining in the figure legend to Figure 1.

*2)*
Figure 2*: The expression of CXCR3 early by activated CD8*^*+*^
*T cell being IL12-independent but Tbx21(T-bet)-dependent is very interesting? Have the authors looked at other factors that drive T-bet, such as IL-27? This would be very interesting to examine. But definitely should be discussed more thoroughly in the manuscript.*

We agree with the reviewer that this is an interesting subject matter. However, our manuscript’s focus in on the mechanisms of CXCR3 downregulation, rather than its upregulation, which is where IL-27 might play a role.

*3)*
Figure 7*: In vivo do IFN-γ receptor deficient CD8*^*+*^
*T cells have altered CXCR3 regulation like is suggested by the in vitro experiments present in this figure. Demonstrate this in vivo would be nice*.

We agree with the reviewer, but this would require crossing our monoclonal (SCNT) T cell mouse into the receptor deficient background and performing adoptive transfer experiments. We plan to do this in the future, but our ex vivo data clearly show that IFN-γ can directly effect CXCR3 downregulation without additional accessory cells.

4) The finding that myeloid DCs express both IL-12 and CXCL9/10 at D3 post-challenge is very interesting. Can the author directly evaluate the importance of this cell population in regulating CXCR3 expression in vivo?

We have performed a new experiment to address this question. Our new data is shown in Figure 7–figure supplement. (Please also see our response to reviewer 2 above.)